# Mutational Analysis of the Binding of Alternative Substrates and Inhibitors to the Active Site of Human Glutathione Transferase P1–1

**Abeer Shokeer [1], Aram Ismail [2], Usama M. Hegazy [1], Rüdiger H. Kolm [2] and Bengt Mannervik [2],***

[1] Department of Molecular Biology, Genetic Engineering and Biotechnology Research Division, National Research Centre, Dokki, 126 22 Giza, Egypt; abeergs70@hotmail.com (A.S.); usama.hegazy@gmail.com (U.M.H.)
[2] Department of Biochemistry and Biophysics, Stockholm University, 106 91 Stockholm, Sweden; Aram.Ismail@dbb.su.se (A.I.); Notapplicable@notapplicable.com (R.H.K.)
* Correspondence: bengt.mannervik@dbb.su.se; Tel.: +46-08-16-18-82

**Abstract:** Glutathione transferases (GSTs) are enzymes that play a critical role in cellular detoxication by catalyzing the nucleophilic attack of glutathione on the electrophilic center of a number of xenobiotic compounds, including many therapeutic drugs. Mutations of amino acid residues in the glutathione-binding site of human glutathione transferase P1–1, namely W39C, K45A, Q52A, Q52K, and Q52E, have been engineered. The recombinant mutant proteins were expressed in *Escherichia coli*, but only mutants K45A, Q52A, and Q52K showed measurable activity. Steady-state kinetics comparing glutathione with the alternative thiol substrate γ-glutamylcysteine demonstrated the importance of the glycine residue in glutathione for high catalytic efficiency. Inhibition experiments with a set of glutathione analogs structurally related to the therapeutic drugs Telintra and Telcyta enabled determination of binding energies that were contributed by different substituents. The effects of substituting amino acid side chains in the glutathione-binding site of the enzyme on binding the glutathione derivatives and catalysis were evaluated.

**Keywords:** Glutathione transferase P1–1; Telintra; Telcyta; alternative thiol substrates; inhibitor binding energy

## 1. Introduction

Human glutathione transferase (GST) P1–1 is a member of a superfamily of detoxication enzymes, the GST superfamily [1,2]. This cytosolic enzyme is overexpressed in many malignant tumors [3,4] and is implicated in resistance to chemotherapy. Inhibition of the enzyme could, therefore, be expected to increase the efficiency of cytostatic drugs that, otherwise, would be inactivated by the enzyme [5]. This notion has been a rationale for the design of GST P1–1 inhibitors for use as adjuvant agents in chemotherapy [6,7].

The GST P1–1 protein consists of two identical subunits of 23 kDa molecular mass. Each subunit can bind a molecule of glutathione (GSH) and a hydrophobic electrophile in two adjacent sites of the catalytically active center, the G-site and the H-site, defined by Mannervik et al. [8]. The three-dimensional structure of the enzyme has been solved [9,10], and the role of individual functional groups can be assessed by kinetic studies of point-mutated variants of hGST P1–1. In earlier publications [11,12], the role of some of the amino acid residues forming the G-site of GST P1–1 has been investigated in terms of their contribution to the kinetic properties. In the present investigation, interactions important for the action of inhibitory GSH derivatives are probed (Figure 1).

**Figure 1.** The two alternative thiol substrates and the competitive inhibitors used in the study.

In particular, the compounds originally designated TER117 (γ-L-glutamyl-S-(benzyl)-L-cysteinyl-R(-)-phenylglycine) and TER286 (γ-glutamyl-α-amino-β-(2-ethyl-N,N,N′,N′-tetrakis (2-chloroethyl)phosphorodiamidate)sulfonylpropionyl-R(-)phenylglycine were of interest. Both substances are S-substituted glutathione molecules in which glycine has been replaced

by R(-)-phenylglycine, and, thereby, affording enhanced selectivity for binding to GST P1–1 rather than to other GSTs [13]. The compounds have therapeutic potential in divergent applications and via different mechanisms. TER117 promotes the proliferation and maturation of hematopoietic precursor cells, granulocytes, monocytes, erythrocytes, and platelets [14,15]. It is administered as the di-esterified molecule called Telintra or Ezatiostat, which, intracellularly, is hydrolyzed to TER117. By contrast, TER286, with the commercial name Telcyta or Canfosfamide, is an anti-proliferative drug of the nitrogen mustard type designed to target GST P1–1-expressing tumors [16,17].

GST P1–1 binds GSH by an array of amino acids residues that form a number of salt bridges and hydrogen bonds. GSH has a carboxyl group at either end of the molecule, providing two negative charges, which are involved in electrostatic interactions with the protein. The γ-glutamyl carboxyl group is surrounded by Ser66 and Argl4 in which the latter was the subject of an earlier investigation [12]. The glycine carboxyl group, on the other hand, is surrounded by Lys45, Gln52, and Trp39, forming a salt bridge and two hydrogen bonds (Figure 2A). TER117 and TER286 both feature R(-)-phenylglycine, and this bulky and stereospecific substitution was of particular interest in the present investigation.

Gln52 and Lys45 were both separately mutated into alanine, known as the most "neutral" of all mutations possible. The alanyl side chain has no marked chemical or electrostatic properties and, at the same time, conserves some of the original backbone rigidity of the native enzyme. The positively charged amino group of the Lys45 side chain matches the negatively charged carboxylate of the glycine residue of the GSH derivatives (Figure 2A–C). The Gln52 side chain amido group is forming a hydrogen bond to the same carboxyl group of the ligand. The Gln52 residue was also mutated into lysine and into glutamic acid, and, thereby, introducing either a positive or a negative charge to the ligand binding site. In addition, Trp39 was mutated into cysteine, which could potentially contribute a hydrogen bond via its sulfhydryl group, even though the donor is further removed than the –NH- of tryptophan. The mutant enzymes were purified and assayed with alternative substrates and inhibitory GSH derivatives.

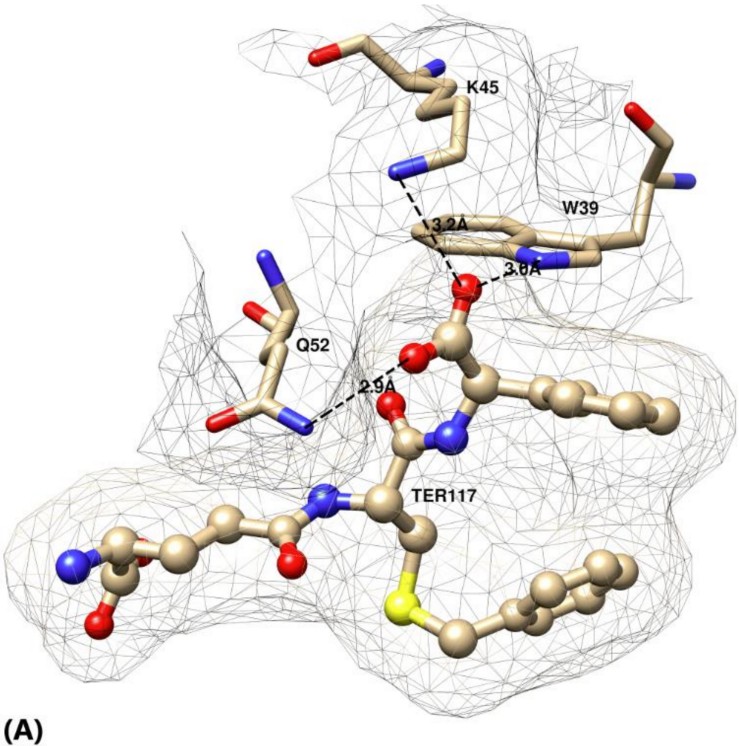

**(A)**

**Figure 2.** *Cont.*

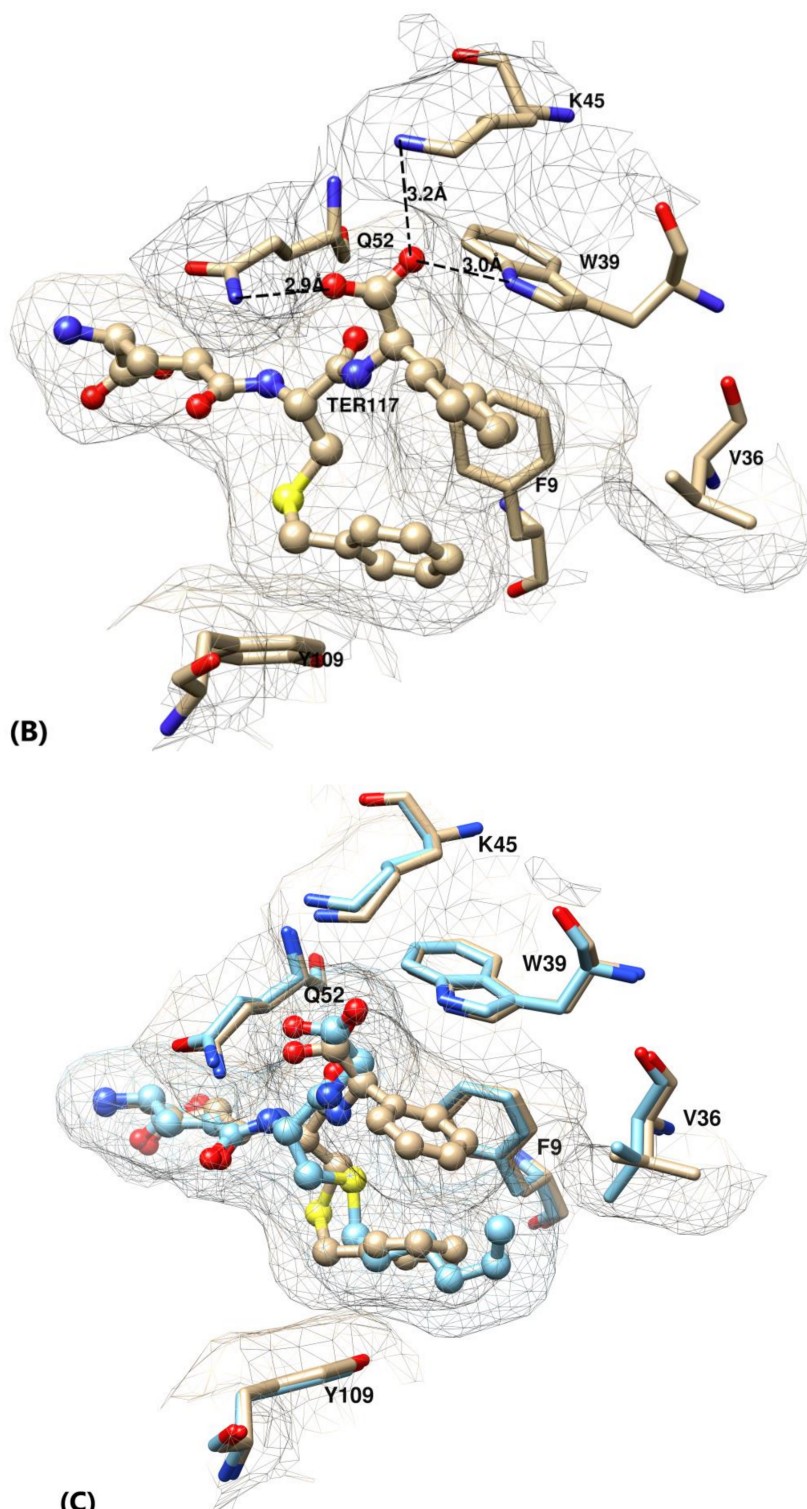

**Figure 2.** (**A**–**C**) Active site of glutathione transferase (GST) P1–1 in complex with TER117 based on crystal structure 10GS. (**A**) Interactions of the C-terminal R(-)-phenylglycine of TER117 with the three mutated residues Trp39, Lys 45, and Gln52. (**B**) The hydrophobic pocket formed by residues Phe9, Val36, and Tyr109 by accommodating the phenyl group of TER117. Note the intramolecular interaction of the S-benzyl substituent of TER117 with the phenyl group. (**C**) Superposition of the structure of GST P1–1 with S-hexylglutathione (light blue, pdb 9GSS) on the TER117 complex showing the similar poses of the S-benzyl and S-hexyl substituents.

## 2. Materials and Methods

### 2.1. Chemicals

Radioactive nucleotides for DNA sequencing were provided by GE Healthcare (GE Healthcare, USA). Enzymes for DNA restriction and modification as well as the phage vector M13mpl8 and *E. coli* strain JM109 were bought from Promega Corp. (Madison, WI, USA). Mutagenesis reagents were purchased as a kit from Amersham (Amersham, Bucks., UK). Sequencing reagents were obtained from Boehringer Mannheim (Mannheim, Germany). Buffer chemicals of the highest quality available were bought from Merck (Darmstadt, Germany). Bacterial broth components were bought from Difco (Detroit, MI, USA). Oligonucleotides were synthesized at Operon Technologies (Alameda, CA, USA). S-Hexylglutathione [$\gamma$E-C(Hx)-G], and S-benzylglutathione [$\gamma$E-C(Bz)-G] were synthesized according to the method of Vince et al. [18]. $\gamma$-Glutamyl-(S-hexyl)-cysteinyl-phenylglycine [$\gamma$E-C(Hx)-ØG], $\gamma$-glutamyl-(S-benzyl)-cysteinyl-phenylglycine [$\gamma$E-C(Bz)-ØG], and $\gamma$-glutamyl-(S-benzyl)-cysteinyl-$\beta$-alanine [$\gamma$E-C(Bz)-$\beta$A] were synthesized at Terrapin Technologies Inc. (San Francisco, CA, USA). For simplicity, the compounds will be named by the abbreviations given. $\gamma$-Glutamylcysteine ($\gamma$E-C) was purchased from Kohjin Co. Ltd., Tokyo, Japan. Glutathione (GSH) and 1-chloro-2,4-dinitrobenzene (CDNB) were from Sigma-Aldrich. All other chemicals were of the highest purity available.

### 2.2. Plasmid and Phage Constructions

The plasmid pPCR Pi [11] was used to produce mutants Q52A and K45A, whereas the plasmid pKXHPl [19] was used for the mutants W39C, Q52K, and Q52E. Both plasmids are based on the expression vector pKK-D [20] and afford heterologous expression of proteins in *E. coli* under the control of the *tac* promoter.

The following oligonucleotide sequences were used: W39C (5′-GAG CCC TCC TGG CAC GTC TCC ACG G−3′), K45A (5′-GCA GGA GGC TGC CGA GTG AGC C−3′), Q52A (5′-CTT GGG GAG CGC CCC GTA TAG−3′), Q52E/K (5′-CTT GGG GAG CT(C/T) CCC GTA TAG GC−3′). The appropriate cDNA-fragment was transferred into M13mpl8. The mutagenesis was carried out according to the method of Taylor et al. [21]. After the mutagenesis reactions, candidate clones were sequenced and the correctness of the entire coding sequence, containing the desired mutation, was verified. The cDNA insert was isolated using standard procedures [22] and transferred into the expression plasmid pKK-D [20]. The plasmid was maintained as a pure DNA preparation, and used for transformation of the *E. coli* JM109 strain.

### 2.3. Expression and Purification of Recombinant Proteins

*E. coli* JM109 containing the appropriate expression plasmid was grown in a rotary shaker as described earlier [19]. If required, the eluted protein was concentrated in an ultrafiltration cell (Amicon) fitted with a 25 Å pore membrane (Spectrum, Houston, TX, USA) with a cutoff $M_r$ of 10,000 kDa. The remaining volume was dialyzed for 24 h against $6 \times 2$ L 100 mM Tris/HCl pH 7.8. The expression of recombinant GST P1–1 protein in crude bacterial lysates was quantified by ELISA as earlier described [4] except for the previously characterized wild-type enzyme and mutant K45A [12], which were quantified by activity measurements.

Protein concentrations were determined by the method of Peterson [23]. Purity and homogeneity were checked on SDS-PAGE (12.5%, *w/v*, acrylamide, 0.5%, *w/v*, N,N′-bisacrylamide) with subsequent silver staining [24]. HPLC analysis of purified proteins [25] was performed on a Dynamax 300A reverse phase C4 column (Rainin Instruments, Woburn, MA, USA). The gradient used was 1 min of 45% (*v/v*) $CH_3CN/H_2O$, 0.1% (*v/v*) trifluoroacetic acid, which was followed by 20 min of 45–60% (*v/v*) $CH_3CN/H_2O$, 0.1% (*v/v*) trifluoroacetic acid.

## 2.4. Kinetic Parameters

Initial-rate kinetics measurements were carried out on a Hewlett-Packard A8452A diode array spectrophotometer according to the conditions specified in Tables 1–3. For initial-rate measurements involving two varied substrate concentrations and for inhibition studies, a 96-well-plate-reader (Molecular Devices, Inc., Menlo Park, CA, USA) fitted with polystyrene plates (Greiner) was used. In order to check for possible adsorption of the enzyme that could interfere with the assay, the plates were coated with buffer containing 1% (*w/v*) bovine serum albumin and 2 % (*v/v*) Triton X−100. However, the rates of the enzyme reactions were not detectably influenced by pre-coating of the plates. Curve fitting and regression analyses were performed using the program SIMFIT [26].

**Table 1.** Expression levels, purification method, and specific activities of glutathione transferase (GST) P1–1 variants.

| GST P1–1 Variant | Expression Level (mg/L Culture) | Affinity Matrix Used for Purification | Specific Activity (μmol/min/mg) [a] |
|---|---|---|---|
| wild-type | 220 [b] | S-hexylglutathione-Sepharose | 110 |
| Q52K | 88 [c] | S-hexylglutathione-Sepharose | 35 |
| Q52A | 5.1 [c] | S-hexylglutathione-Sepharose | 23 |
| Q52E | 7.9 [c] | glutathione-Sepharose | <0.1 |
| K45A | 1.4 [b] | S-hexylglutathione-Sepharose | 32 |
| W39C | 1.8 [c] | not purified | <0.05 |

[a] 1 mM Glutathione, 1 mM CDNB, 30 °C, 100 mM sodium phosphate pH 6.5. [b] Based on activity measurements in bacterial lysate. [c] Value determined by ELISA.

**Table 2.** Steady-state kinetic parameters of wild-type human GST P1–1 and mutants.

| Parameter | Varied Substrate | GST P1–1 Variant | | | |
|---|---|---|---|---|---|
| | | Wild-Type | Q52K | Q52A | K45A |
| $k_{cat}/K_m$ (s$^{-1}$ mM$^{-1}$) | CDNB | 83 ± 2 | 4.9 ± 0.5 | 25.6 ± 0.9 | 46 ± 22 |
| $k_{cat}/K_m$ (s$^{-1}$ mM$^{-1}$) | Glutathione | 345 ± 9 | 5.0 ± 0.5 | 17.3 ± 0.8 | 2.2 ± 0.4 |
| $K_m$ (mM) | CDNB | 2.6 + 0.2 | 7.0 ± 2 | 6.2 ± 0.8 | 3.8 ± 0.4 |
| $K_m$ (mM) | Glutathione | 0.5 ± 0.1 | 4.7 ± 0.6 | 2.8 ± 0.3 | 0.8 ± 0.1 |
| $V_{max}$ (μmol min$^{-1}$ mg$^{-1}$) | | 570 ± 30 | 70 ± 20 | 340 ± 30 | 440 ± 30 |

One substrate concentration was varied, whereas the concentration of the other substrate was kept constant. Constant CDNB was kept at 2 mM, and constant glutathione at 6 mM in order to maintain approximately saturating conditions.

**Table 3.** Saturation kinetics in three dimensions.

| $V_{max}$ (μmol min$^{-1}$ mg$^{-1}$) | | | | | |
|---|---|---|---|---|---|
| [CDNB] (mM) | Wild-Type GSH | Q52K GSH | Wild-Type γE-C | Q52K γE-C |
| 0.2 | 63 | 113 | 7.0 | <0.1 |
| 0.5 | 110 | 291 | 9.9 | 0.6 |
| 1 | 162 | 592 | 11.6 | 4.3 |
| "Saturated" | 278 | No value | 13.8 | No value |

$V_{max}$ values as obtained for three different concentrations of CDNB with variable concentrations of glutathione (GSH) or γ-glutamylcysteine (γE-C).

Inhibition constants were determined from inhibition studies for all enzymatically active mutants. Three inhibition experiments were done using three different concentrations of GSH (0.2 mM, 0.5 mM, 1.0 mM) and a constant concentration of CDNB (1 mM). The assay was performed at 30 °C in 100 mM sodium phosphate at pH 6.5, containing 2% (*v/v*) ethanol. IC$_{50}$ values were calculated as the concentration of the inhibitor that gives 50% of the reaction rate of the uninhibited reaction. Standard deviations of the measured parameters were less than 10%.

*2.5. Structural Models of Enzyme-Inhibitor Complexes*

Active-site models of GST P1–1 in complex with TER117 and S-hexylglutathione were depicted using the software UCSF Chimera (http://www.rbvi.ucsf.edu/chimera) and were based on the Protein Data Bank structures 10GS and 9GSS, respectively.

## 3. Results and Discussion

*3.1. Expression of GST P1–1 Variants*

Wild-type human GST P1–1 as well as mutant enzymes W39C, K45A, Q52A, Q52E, and Q52K were expressed in *E. coli*. The presence of mutant proteins in the bacterial lysate was verified and quantified by ELISA (Table 1). In particular, it was shown that mutant W39C was expressed. However, the lysate had virtually no catalytic activity with CDNB, and the mutant enzyme could not be obtained as a pure protein. Mutants K45A, Q52A, and Q52K were purified on an S-hexylglutathione affinity column. Q52E did not bind to this column and was purified on a GSH affinity matrix. The homogeneity of the four purified enzymes was verified by HPLC and SDS-PAGE.

*3.2. Specific Activities of hGST P1–1 Variants*

Specific activities were determined (Table 1), and the values obtained for mutants Q52K, Q52A, and K45A were 20–30% of the wild-type activity, suggesting that the mutations do not involve residues essential for catalytic activity. On the other hand, mutants Q52E and W39C did not display any measurable activity ($<10^{-3}$ of the specific activity of the wild-type GST P1–1). The poor binding of mutant Q52E to the S-hexylglutathione affinity matrix may be related to the lack of catalytically productive substrate binding, presumably due to repulsion of the negatively GSH molecule caused by the introduction of a carboxylate group in the enzyme. The lack of significant activity of mutant W39C may be due to structural rearrangements in the active site, caused by removal of the bulky and hydrophobic indole ring of W39, which is located in a crucial position in the α-helix 2 [9]. It is also possible that the protein is not properly folded, even though its production was verified by ELISA.

Mutant Q52E has negligible catalytic activity in comparison to mutants Q52K and Q52A. This result is in agreement with the role of Q52 in binding the C-terminal carboxyl group of the GSH molecule. A slight dislocation or a loss of the Q52 hydrogen bond in Q52K and Q52A is compatible with binding of GSH, whereas the repulsion caused by the carboxyl group in Q52E prevents binding.

*3.3. Steady-State Kinetics*

Initial rates were measured with wild-type GST P1–1, Q52K, Q52A, and K45A using GSH and CDNB as substrates (Table 2). The analysis is somewhat limited by the difficulties in reaching rate-saturating concentrations of CDNB due to its limited solubility. Nevertheless, it is clear that effects of the mutations are expressed mainly in increased $K_M^{GSH}$ and decreased $k_{cat}$ values, whereas $K_M^{CDNB}$ is only moderately increased. Mutant Q52K is most strongly affected with a nine-fold increase of $K_M^{CDNB}$, an eight-fold decrease of $k_{cat}$, and a resulting 70-fold decrease of $k_{cat}/K_M^{GSH}$ (Table 2). The 20-fold and 16-fold decreases of $k_{cat}/K_M^{GSH}$ for mutants Q52A and K45A, respectively, are primarily due to increased $K_M$ values.

GSH and γ-glutamylcysteine (γE-C) were used to probe the role in the catalytic mechanism of the glycine residue in the thiol substrate. The dipeptide γE-C is the biosynthetic precursor of GSH and is present in low concentrations in the cell. Wild-type GST P1–1 and mutant Q52K were assayed with different concentrations of CDNB (Table 3). The resulting apparent $V_{max}$ values were in their turn fitted with the Michaelis-Menten equation in a secondary regression analysis to give an extrapolated reaction rate valid for infinite concentrations of both substrates. Mutant Q52K did not show saturation in the experimentally available concentration range.

However, the catalytic efficiency could be determined and showed that mutant Q52K was 50% more efficient than the wild-type enzyme with GSH as the varied thiol substrate. Apparently,

the positively charged lysine side-chain stabilizes the transition state of the reaction involving GSH. The alternative thiol substrate $\gamma$E-C showed 18% of the catalytic efficiency with the wild-type enzyme. By contrast, mutant Q52K tested with $\gamma$E-C gave only 5% of the wild-type activity with the same alternative thiol substrate (Table 4), indicating loss of transition-state stabilization. Possibly, the ionic interaction between the lysyl residue and the cysteinyl carboxylate is shifting the position of the thiol substrate to a less favorable position for the nucleophilic attack on the electrophilic substrate CDNB.

**Table 4.** Catalytic efficiency ($k_{cat}/K_m$) of GST P1–1 variants with alternative thiol substrates.

| | GST P1–1 Variant | |
| --- | --- | --- |
| **Substrate** | **Wild-Type** $k_{cat}/K_m$ **(mM$^{-1}$ s$^{-1}$)** | **Q52K** $k_{cat}/K_m$ **(mM$^{-1}$ s$^{-1}$)** |
| GSH | $380 \pm 40$ | $590 \pm 6$ |
| $\gamma$E-C | $70 \pm 1.3$ | $3.6 \pm 1.4$ |

*3.4. Inhibition Studies*

Inhibition studies with a series of S-substituted GSH analogs were performed with GSH and CDNB as substrates. Two inhibitors carried a hexyl group on the sulfur, two contained phenylglycine instead of glycine in the C-terminus of the inhibitory peptide, three carried a benzyl group on the sulfur, and one contained β-alanine instead of glycine (Figure 1). The inhibitors were competitive with GSH, and the IC$_{50}$ values determined for different GSH concentrations were extrapolated to zero substrate concentration to give the inhibition constants [27]. The corresponding incremental binding energies ($\Delta\Delta G$) were calculated as

$$\Delta\Delta G \ = \ RT \ \ln \ K_i K_i^*) \tag{1}$$

where $K_i^*$ is the inhibition constant for wild-type enzyme inhibited with S-hexylglutathione, which was used as a reference (Table 5). The relationships among the $\Delta\Delta G$ values have been arranged into a thermodynamic cycle, which demonstrates additivity of the alterations and, thus, internal consistency of the data (Figure 3). As the enzyme reference, the wild-type variant was chosen for clear reasons, and, in the case of the ligands, $\gamma$E-C(Hx)-G, since this is an inhibitor with a defined binding mode in the H-site [9,10]. The additivity signifies that no major secondary changes overrule the primary effects of the changes in structure of the enzyme as well as inhibitor. The calculated energy values, expressed as differences between the reference set and each particular mutant with the five GSH analogs, allow the estimation of the influence of particular side-chain alterations in the G-site of the substrate as well as in the bound ligand (Table 5).

$\gamma$E-C(Bz)-ØG is the strongest binder in the wild-type enzyme as well as in all the mutants tested. Since TER117 outcompeted $\gamma$E-C(Bz)-G in all cases, interaction of the phenyl substituent on the glycine residue with the enzyme is indicated. The crystal structure of the GST P1–1 complex with TER117 [10] shows that the phenyl group of the glycine residue is positioned in a pocket formed by the side-chains of Phe9, Val36, and Trp39 (Figure 2B). The gain in binding energy afforded by the phenyl group is 8.0 kJ mol$^{-1}$ (Figure 3). The phenyl group also interacts with the benzyl group of TER117 [10]. This intramolecular interaction is substantial, as shown by comparison with the S-hexyl substituted $\gamma$E-C(Hx)-G and $\gamma$E-C(Hx)-φG. Substituting benzyl for hexyl in $\gamma$E-C(Hx)-φG leads to an improved binding energy of 10.7 kJ mol$^{-1}$ (Figure 3). The superposition of the structures of the complexes of wild-type enzyme with $\gamma$E-C(Hx)-G and with TER117 shows that the hexyl and benzyl groups are located in essentially the same position (Figure 2C). Apparently, the stacking of the benzyl substituent of TER117 between the aromatic side-chains of Phe9 and Tyr109 provides additional binding energy. This effect is shown in the mutants K45A, Q52A, and Q52K, but to a more limited extent (2.2–3.9 kJ mol$^{-1}$, Figure 3).

All mutants showed decreased binding affinities for the five GSH derivatives (Table 5), as could be expected considering the loss of binding functions of the mutated residues. The most detrimental

mutation was Q52K, which is caused by repulsion between the positive charge of the new side-chain in residue 52 and the ammonium group of K45.

$\gamma$E-C(Bz)-$\beta$A is the S-benzyl derivative of homoglutathione, which is naturally occurring in plants. This compound is the weakest inhibitor with all enzyme variants investigated (Table 5). The positioning of the carboxyl group one methylene group away from the location of the carboxyl group of the glycine residue in GSH results in a loss of binding energy of 7.8 kJ mol$^{-1}$ in the wild-type GST P1–1 and 6.8 kJ mol$^{-1}$ in mutant Q52A. This minor structural difference gives rise to a major loss of binding energy as demonstrated by the values for $\gamma$E-C(Bz)-G and $\gamma$E-C(Bz)-$\beta$A obtained with all GST P1–1 variants (Table 5).

The inhibition constants and, consequently, the binding energies for both $\gamma$E-C(Hx)-G and $\gamma$E-C(Bz)-G are of similar magnitude, which makes it possible to use them in parallel to dissect effects in the G-site with simultaneous changes in the H-site. For these two GSH conjugates, which are possible products of enzymatic GSH conjugation of, e.g., hexyl and benzyl halides, all three mutant enzymes show a loss in affinity in comparison with wild-type GST P1–1. For $\gamma$E-C(Hx)-G, this loss is of the same order of magnitude with either of the three mutants investigated with an average around 6 kJ mol$^{-1}$.

Mutant Q52K shows weaker binding of all ligands than shown by mutant Q52A, indicating that introduction of the longer and positively charged side-chain of a lysine residue is more damaging than the total loss of interactions originating from this position in the amino acid sequence. With $\gamma$E-C(Bz)-G, mutant Q52A is the best binder following the wild-type enzyme, indicating that the absence of the Q52 side chain is less detrimental than the Q52K replacement. This consequence of the Q52 replacement is maintained among all the ligands tested with the loss of binding energy being 4–5 kJ mol$^{-1}$. $\gamma$E-C(Bz)-$\beta$A displays this trend, even though this ligand with an extra methylene group in the C-terminus deviates somewhat from the shape of GSH.

$\gamma$E-C(Hx)-ØG is an interesting analog to yE-C(Hx)-G, as it does not follow the trend of values observed for $\gamma$E-C(Bz)-$\beta$A. Especially for the K45A mutation, no significant loss of binding strength is observed, which is consistent with the structural data showing that the phenyl group fits into a space. This is left intact by the K45A mutation [10]. Notably, the loss of binding energy of both mutants Q52A and K45A are similar to each other for the five ligands tested (Table 5). The loss of the positive charge in residue 45 is not generally more harmful than the loss of a hydrogen bond from residue 52.

**Table 5.** Inhibitions constants and incremental binding energies as determined from inhibition studies with glutathione analogs.

| Ligand | GST P1–1 Variant | | | |
|---|---|---|---|---|
| | Wild-Type | Q52K | Q52A | K45A |
| **$\gamma$E-C(Hx)-G** | | | | |
| Ki ($\mu$M) | 4.2 | 70 | 65 | 35 |
| $\Delta\Delta$G (kJ mol$^{-1}$) | 0 | −7.0 | −6.9 | −5.3 |
| **$\gamma$E-C(Hx)-$\Phi$G** | | | | |
| Ki ($\mu$M) | 20 | 270 | 45 | 25 |
| $\Delta\Delta$G (kJ mol$^{-1}$) | −3.9 | −10.5 | −6.0 | −4.5 |
| **$\gamma$E-C(Bz)-$\Phi$G** | | | | |
| Ki ($\mu$M) | 0.3 | 40 | 7 | 8.5 |
| $\Delta\Delta$G (kJ mol$^{-1}$) | 6.7 | −5.7 | −1.3 | −1.8 |
| **$\gamma$E-C(Bz)-G** | | | | |
| Ki ($\mu$M) | 8.3 | 150 | 17 | 40 |
| $\Delta\Delta$G (kJ mol$^{-1}$) | −1.7 | −9.0 | −3.5 | −5.7 |
| **$\gamma$E-C(Bz)-$\beta$A** | | | | |
| Ki ($\mu$M) | 180 | 550 | 250 | 170 |
| $\Delta\Delta$G (kJ mol$^{-1}$) | −9.5 | −12.5 | −10.3 | −9.3 |

Binding energies were calculated using GST P1–1 wild-type enzyme with S-hexylglutathione as a reference. Negative values signify decreased binding affinities.

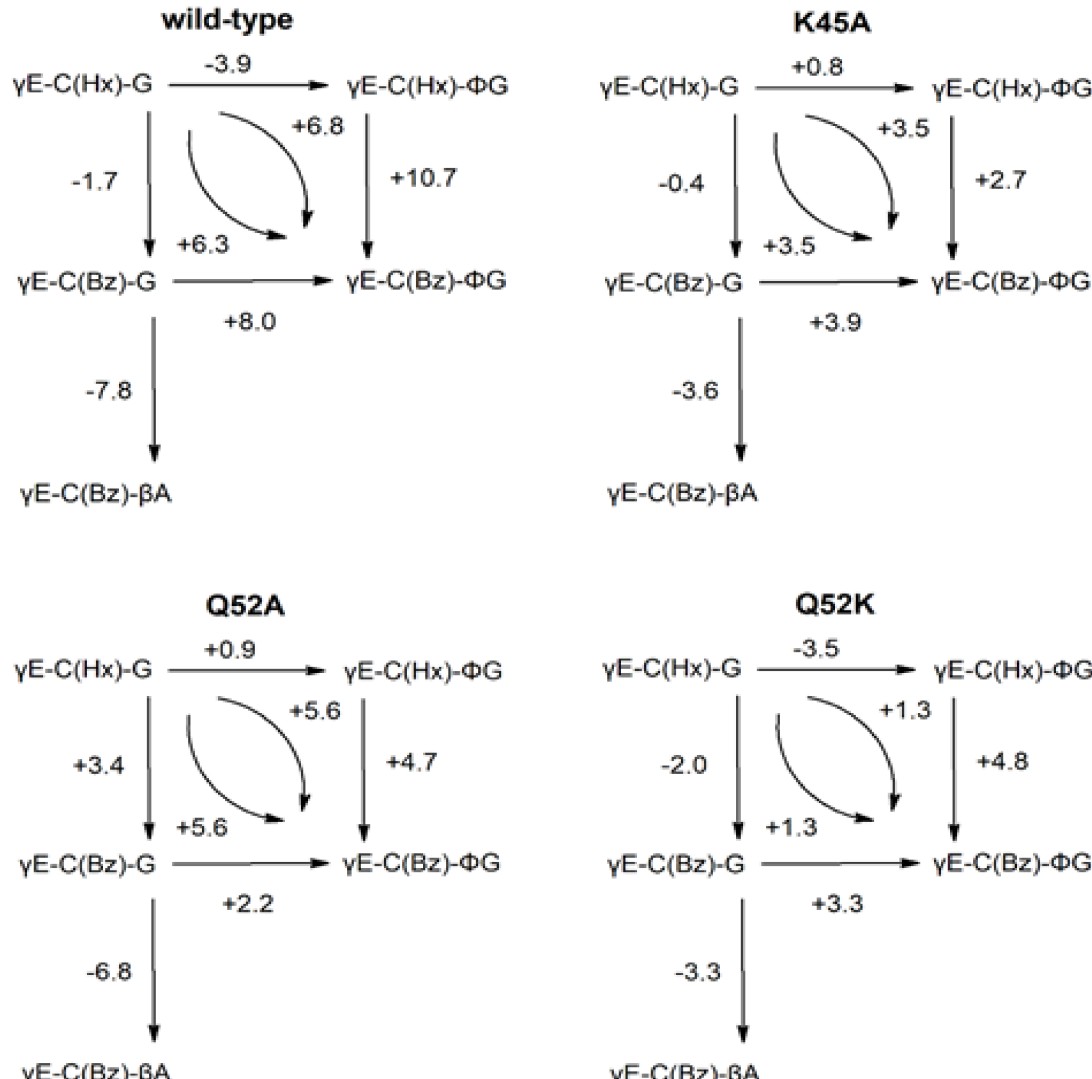

**Figure 3.** Schemes of incremental binding energies (kJ mol$^{-1}$) in comparison of the binding of five different glutathione derivatives to wild-type GST P1–1 and mutants K45A, Q52A, and Q52K. The complexes with S-hexylglutathione, $\gamma$E-C(Hx)-G (upper left corners) serve as references. Positive values indicate increased binding affinities.

The benzyl analog of the above ligand, $\gamma$E-C(Bz)-$\emptyset$G (TER117), shows a different picture: mutants Q52A and K45A both cause a loss in binding strength of approximately 8 kJ mol$^{-1}$, reflecting the binding energy characterizing each of these two separate interactions. Q52K gives an additional loss of 4.4 kJ mol$^{-1}$, which corresponds to the result obtained with $\gamma$E-C(Bz)-$\emptyset$G. With $\gamma$E-C(Bz)-$\emptyset$G, the Q52K mutation is, therefore, the most consequential one, giving the highest absolute loss of binding energy ($-12.4$ kJ mol$^{-1}$) of all values (Table 5). On the other hand, the same ligand is found to be a potent ligand with wild-type enzyme GST P1–1, indicating a role of residue Q52 in positioning the phenyl group in the hydrophobic binding cavity in the G-site of the enzyme (Figure 2B).

TER117 and TER286, which are both of therapeutic value, share the structure of the modified GSH scaffold including the R(-)-phenylglycine (Figure 1). This feature is instrumental in securing the selective binding to GST P1–1 in comparison with other soluble GSTs [13], since steric clashes in the active site appear to impede binding to mu and alpha class GSTs [10]. As noted above, the interaction of the phenyl group with the benzyl substituent in TER117 makes a significant contribution to the high affinity of GST P1–1 for the compound. A corresponding interaction between the aromatic group cannot take place in TER286, which has a polar phosphorodiamide mustard linked to an oxidized

sulfur. TER286 was not available for the present investigation, and the crystal structure of its complex with GST P1–1 has not been determined. It remains to find out if TER286 has a binding energy more similar to that of $\gamma$E-C(Hx)-$\emptyset$G than TER117.

## 4. Conclusions

The present investigation demonstrates the useful combination of structural variations of amino acid residues in the binding site of a protein with different substituents in the bound ligand. Determination of the binding energies for the different combinations contributes to the detailed elucidation of the interactions important in the interplay between structure and activity.

**Author Contributions:** A.S., A.I, U.M.H., R.H.K., and B.M. designed the experiment, conducted laboratory analyses, wrote parts of the manuscript, and interpreted the results. A.S., A.I., U.M.H., and B.M., visualized and revised the article. All authors have read and agreed to the published version of the manuscript.

**Funding:** The Swedish Natural Science Research Council (grant 2015-04222), the Swedish Childhood Cancer Fund (PR 2019–0116), and the Carl Trygger Foundation funded this research.

**Acknowledgments:** The glutathione analogs with modified glycine moieties were synthesized and kindly provided by Matthew, H. Lyttle, Terrapin Technologies, Inc., S. San Francisco, CA, USA. We thank B. Olin and X.-Y. Hao for excellent help with the HPLC and ELISA quantitation.

**Conflicts of Interest:** The authors declare no conflict of interest.

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
