# Peer review of "Mutational Analysis of the Binding of Alternative Substrates and Inhibitors to the Active Site of Human Glutathione Transferase P1–1"

_processes, doi:10.3390/pr8101232_

Round 1

Reviewer 1 Report

In this manuscript, the authors investigate mutants within the active site of human GST PI-1, with hopes of determining which residues influence the binding of various substrates and inhibitors.  The residues selected for mutation (W39, K45, and Q52) were principally located in the pocket that binds the carboxylate of GSH’s glycine moiety.  The authors found that residues in this site – namely K45 and Q52 – are critical for binding interactions and efficacy of various inhibitors of the enzyme and that these residues exert this effect by facilitating GSH binding.  The manuscript, while somewhat limited in scope, is fairly well-written and straightforward.  I have a few points for the authors to consider in a revision.  If these are adequately addressed, I feel the manuscript will be appropriate for publication. 

My main concern deals with why the authors selected to make the mutations of K45 and W39 that they did.  In particular, I am unclear on why authors made the W39C mutant.  It did not have any appreciable activity and could not be purified.  Why not focus on W39F or W39Y or even W39I or W39L instead? Likewise, I am unclear on why the authors only studied the K45A mutant without studying K45Q and K45E, allowing them to do the corresponding mutagenesis to what they did with Q52?  The choice of mutants should be more clearly justified somewhere in the text, either in the introduction or in the beginning part of the results section to make clear the rationale for making each of these mutants.  The rationale for making the Q52 mutants is clear, just not the others.  Along these lines, were any combination mutants – perhaps at the K45 and Q52 sites – ever studied to see if they could counteract one another?

Typographical errors/minor corrections

  • The title on lines 2-3 continues onto line 4 in a smaller font point.
  • Line 59: TER286 rather than TER 286.
  • Line 90: S-Hexylglutathione rather than S-Hexylglutaihione.
  • Line 121: Indicate what the analyte is in the sentence that begins “HPLC analysis (of ?) [25]”
  • Line 128: 96-well plate looks like it has a capital “i” in it

Reviewer 2 Report

Main comments

In this manuscript the authors performed mutations in 5 aminoacid residues in the glutathione-binding site of human glutathione transferase Pl-1. They determined the inhibition constants and binding energies for the different ligands and how specific mutations can contributes to significant modifications on the structure and activity of glutathione transferase Pl-1.

This is a very interesting study that could greatly benefit from some computational studies, namely molecular docking, which would allow a better discussion of the obtained results . From my point of view this is a weakness of the manuscript.

More detailed comments:

P2, Line 27: family >> superfamily

Page 2, Line 44 TER289 or TER286 ???

Page 3. line 58: “The glycine carboxyl group, on the other hand, is surrounded by Lys45, Gln52, and Trp39, forming a salt bridge and a hydrogen bond (Figure 2A). “

The authors only mention 2 interactions, but in the figure we can see 3. This point should be clarified. The interactions should also be marked with a dash line as they are not covalent bonds and would be helpful if their type was identified in the figure. The representation of the interaction distances between the compound TER117 and the aminoacids would also be a positive added value.

Page 5: “Gln52, Lys45, and Trp39 were each separately mutated into alanine” - This is not in accordance with the mutations described in the abstract. This should be corrected.

Page 7, section 3.2: In this section, some docking studies would allow a more detailed understanding of the role of mutated amino acids and the interactions that they establish regarding the protein activity.

Page 8, section 3.3: Once again, in addition to the determination of the inhibition constant, undertaking of docking studies would improve the discussion of the inhibition results obtained by the different ligands in the wild-type and mutant proteins.

Reviewer 3 Report

The manuscript entitled "Mutational analysis of the binding of alternative substrates and inhibitors to the active site of human GSTP1-1" by Shokeer et al has focused on the the different effects of enzymatic activity for amino acid change in active site of GSTP1-1, which is interesting to the field. Whereas the manuscript is very comprehensive and well written, further ammendments are needed prior publication in the Processes journal.

  1. The author need to show the wetern blot or Coomassie gel image for various mutants of human GSTP1-1 recombinant protein after purification from bacteria
  2. In figure 2, how to make the structural model of GSTP1-1 active site? Please include statements in the method section. 

Round 2

Reviewer 2 Report

No suggestions.

Reviewer 3 Report

All my critisims are adequately addressed.